# ACTOR-CRITIC WITHOUT ACTOR

## ABSTRACT

Actor-critic methods constitute a central paradigm in reinforcement learning (RL), coupling policy evaluation with policy improvement. While effective across many domains, these methods rely on separate actor and critic networks, which makes training vulnerable to architectural decisions and hyperparameter tuning. Such complexity limits their scalability in settings that require large function approximators. Recently, diffusion models have recently been proposed as expressive policies that capture multi-modal behaviors and improve exploration, but they introduce additional design choices and computational burdens, hindering efficient deployment. We introduce Actor-Critic without Actor (ACA), a lightweight framework that eliminates the explicit actor network and instead generates actions directly from the gradient field of a noise-level critic. This design removes the algorithmic and computational overhead of actor training while keeping policy improvement tightly aligned with the critic's latest value estimates. Moreover, ACA retains the ability to capture diverse, multi-modal behaviors without relying on diffusion-based actors, combining simplicity with expressiveness. Through extensive experiments on standard online RL benchmarks, ACA achieves more favorable learning curves and competitive performance compared to both standard actor-critic and state-of-the-art diffusion-based methods, providing a simple yet powerful solution for online RL.

## 1 INTRODUCTION

Actor-critic methods represent a foundational paradigm in reinforcement learning (RL), in which a critic estimates action values under the current policy and an actor updates the policy toward higher-value actions (Sutton et al., 1998; Konda & Tsitsiklis, 1999). This alternating cycle of evaluation and improvement is theoretically grounded and has demonstrated strong empirical success across diverse domains (Mnih et al., 2016; Lowe et al., 2017; Haarnoja et al., 2018a;b; Espeholt et al., 2018). However, the alternating updates increase algorithmic complexity, requiring careful tuning of network architectures and learning rates for stability (Andrychowicz et al., 2021), while doubling computation and memory demands, making actor-critic methods less attractive in domains requiring large function approximators (Ouyang et al., 2022; Rafailov et al., 2024). Moreover, the gradual policy updates required by the actor contrast with $Q$-learning's direct maximization of the critic, resulting in slower policy improvement as the actor cannot instantly incorporate the critic's latest estimates.

Recent advances have introduced diffusion models as powerful policy parameterizations for RL (Wang et al., 2022; Chen et al., 2022; Lu et al., 2023; Chen et al., 2023; 2024; Zhu et al., 2024; Zhang et al., 2024; Ren et al., 2024; Lu et al., 2025). These models generate actions by progressively denoising Gaussian noise over a sequence of timesteps, enabling expressive multi-modal action distributions well-suited for complex control. In the offline setting, this expressivity enables diffusion policies to recover high-return trajectories from heterogeneous datasets and surpass unimodal Gaussian policies. This benefit extends to online RL as well, where diffusion policies promote broader exploration and better mode coverage (Wang et al., 2024; Yang et al., 2023; Psenka et al., 2023; Ding et al., 2024).

Although diffusion-based policies provide strong expressivity for modeling complex, multi-modal action distributions, their deployment in online RL introduces substantial practical challenges. In particular, they rely on large denoising networks that significantly increase memory consumption and training time, and often require additional approximations that introduce bias into policy updates (Ma et al., 2025). These factors complicate implementation, exacerbate computational overhead, and ultimately limit scalability in settings where efficient and lightweight adaptation is crucial.

Figure 1: Comparison between standard actor-critic methods and the proposed Actor-Critic without Actor (ACA). Standard methods maintain both an actor and a critic, adding complexity and overhead, whereas ACA eliminates the actor and achieves policy improvement via critic-guided denoising.

To address these limitations, we introduce **Actor-Critic without Actor (ACA)**, a lightweight framework that eliminates the explicit actor network and relies solely on the critic. Inspired by guidance techniques in diffusion-based offline RL, ACA reformulates action sampling as a critic-guided denoising process, where actions are obtained directly from the gradient field of a noise-level critic. A key distinction of ACA is that it preserves the multi-modal behavior inherent to diffusion models without requiring a separately parameterized, computationally heavy actor. Removing the actor not only reduces model complexity but also ensures that diverse action modes are faithfully represented through the critic alone. Moreover, this mechanism replaces standard policy improvement with gradient-based refinement, keeping sampled actions remain aligned with up-to-date value estimates and thereby eliminating the policy lag. Through this design, ACA achieves improved learning curves and competitive performance on MuJoCo continuous control benchmarks compared to both standard actor-critic methods and diffusion-based approaches, while requiring substantially fewer parameters.

## 2 PRELIMINARIES

**Reinforcement learning (RL)**   We consider the RL problem under a Markov Decision Process (MDP) $\mathcal{M} = \{\mathcal{S}, \mathcal{A}, P, r, \gamma, p_0\}$, where $\mathcal{S}$ is the state space, $\mathcal{A}$ is the action space, $P(\mathbf{s}'|\mathbf{s}, \mathbf{a})$ is the transition probability, $r(\mathbf{s}, \mathbf{a})$ is the reward, $p_0$ is the initial-state distribution, and $\gamma \in [0, 1)$ is the discount factor. The goal of RL is to learn a policy that maximizes the expected cumulative discounted reward in MDP. For a policy $\pi(\mathbf{a}|\mathbf{s})$, the state-action value is defined as $Q^\pi(\mathbf{s}, \mathbf{a}) = \mathbb{E}\left[\sum_{\tau=0}^\infty \gamma^\tau r(\mathbf{s}_\tau, \mathbf{a}_\tau)|\mathbf{s}_0 = \mathbf{s}, \mathbf{a}_0 = \mathbf{a}, \pi, P\right]$, where $\tau$ denotes environment timesteps in RL. A central principle of RL is the policy iteration framework, which alternates between two steps. First, *policy evaluation* estimates $Q^\pi$ for a fixed policy, typically by iterating the Bellman operator

$$(\mathcal{T}^\pi Q)(\mathbf{s}, \mathbf{a}) = r(\mathbf{s}, \mathbf{a}) + \gamma \mathbb{E}_{\mathbf{s}' \sim P(\cdot|\mathbf{s}, \mathbf{a}), \mathbf{a}' \sim \pi(\cdot|\mathbf{s}')} [Q(\mathbf{s}', \mathbf{a}')]. \tag{1}$$

Second, *policy improvement* updates $\pi$ toward actions that maximize the expected $Q$-value, e.g., $\pi(\cdot|\mathbf{s}) \leftarrow \arg\max_\mathbf{a} Q(\mathbf{s}, \mathbf{a})$. Actor-critic methods instantiate this paradigm in a parametric form: the critic approximates $Q^\pi$ via Bellman backups, while the actor is updated using the critic's value estimates, thereby coupling policy evaluation and improvement in a single learning loop.

**Denoising diffusion probabilistic models (DDPMs)**   DDPMs (Ho et al., 2020) are a class of generative models that construct samples through a Markov forward-reverse process. In the forward process, Gaussian noise is incrementally added to a clean data sample $\mathbf{x}_0$ over $T$ timesteps, according to $q(\mathbf{x}_t|\mathbf{x}_{t-1}) := \mathcal{N}(\mathbf{x}_t; \sqrt{1 - \beta_t}\mathbf{x}_{t-1}, \beta_t\mathbf{I})$, where $\beta_t \in (0, 1)$ is a predefined variance schedule and $t$ denotes the diffusion timestep. Importantly, this process admits a closed-form expression for sampling $\mathbf{x}_t$ from $\mathbf{x}_0$ at any timestep $t$, $q(\mathbf{x}_t|\mathbf{x}_0) := \mathcal{N}(\mathbf{x}_t; \sqrt{\bar{\alpha}_t}\mathbf{x}_0, (1 - \bar{\alpha}_t)\mathbf{I})$, with $\alpha_t := 1 - \beta_t$ and $\bar{\alpha}_t := \prod_{s=1}^t \alpha_s$. The reverse process starts from standard Gaussian noise $\mathbf{x}_T \sim \mathcal{N}(\mathbf{0}, \mathbf{I})$ and progressively denoises through a parameterized Markov chain:

$$\mathbf{x}_{t-1} = \frac{1}{\sqrt{\alpha_t}} \left(\mathbf{x}_t - \frac{\beta_t}{\sqrt{1 - \bar{\alpha}_t}}\epsilon_\theta(\mathbf{x}_t, t)\right) + \sigma_t\mathbf{z}, \quad \text{where} \quad \mathbf{z} \sim \mathcal{N}(\mathbf{0}, \mathbf{I}), \tag{2}$$

where the variance is fixed as $\sigma_t^2 = \frac{1 - \bar{\alpha}_{t-1}}{1 - \bar{\alpha}_t}\beta_t$. The diffusion model $\epsilon_\theta$ is trained to approximate the added noise by minimizing a simplified surrogate objective derived from a variational bound:

$$\mathbb{E}_{\mathbf{x}_0 \sim \mathcal{B}, \epsilon \sim \mathcal{N}(\mathbf{0}, \mathbf{I}), t \sim \mathcal{U}[1, T]} \left[\left\|\epsilon - \epsilon_\theta\left(\sqrt{\bar{\alpha}_t}\mathbf{x}_0 + \sqrt{1 - \bar{\alpha}_t}\epsilon, t\right)\right\|^2\right]. \tag{3}$$

**Classifier guidance** Diffusion models incorporate guidance mechanisms that bias the generative process toward samples aligned with a desired label (Ho & Salimans, 2022; Dhariwal & Nichol, 2021). Classifier guidance (Dhariwal & Nichol, 2021) trains a noise-level classifier $p_\phi(y|\mathbf{x}_t, t)$ to predict labels from noisy inputs $\mathbf{x}_t$, and its gradient $\nabla_{\mathbf{x}_t} \log p_\phi(y|\mathbf{x}_t, t)$ is used to steer the diffusion sampling process toward the target class $y$. The guided noise prediction is defined as $\hat{\epsilon}(\mathbf{x}_t, t) := \epsilon_\theta(\mathbf{x}_t, t) - w\sigma_t \nabla_{\mathbf{x}_t} \log p_\phi(y|\mathbf{x}_t, t)$, where $w > 0$ denotes a guidance weight. Here, $\epsilon_\theta$ is the noise-prediction network of the diffusion model, and $\hat{\epsilon}$ is the guided variant incorporating classifier gradients. Since diffusion models can be viewed as score estimators (Song et al., 2020), with $\nabla_{\mathbf{x}_t} \log p_t(\mathbf{x}_t) = -\epsilon^*(\mathbf{x}_t, t)/\sigma_t \approx -\epsilon_\theta(\mathbf{x}_t, t)/\sigma_t$, classifier guidance can be reformulated from the perspective of score functions:

$$\nabla_{\mathbf{x}_t} \log \hat{p}_t(\mathbf{x}_t) = \nabla_{\mathbf{x}_t} \log p_{t,\theta}(\mathbf{x}_t) + w\nabla_{\mathbf{x}_t} \log p_\phi(y|\mathbf{x}_t, t) \tag{4}$$

## 3 METHOD

### 3.1 CLASSIFIER-GUIDANCE IN ONLINE RL

Diffusion models have recently been adopted in offline RL as a natural framework for capturing the multi-modal structure of action distributions and mitigating out-of-distribution issues (Wang et al., 2022; Chen et al., 2022; Kang et al., 2023). Among various approaches, diffusion guidance methods have proven effective in directing behavior-cloned diffusion models toward high-return actions (Janner et al., 2022; Ajay et al., 2022; Lu et al., 2023; 2025; Frans et al., 2025). Specifically, in classifier guidance, the score function for a noisy action $\mathbf{a}_t$ at diffusion step $t$ is refined as follows:

$$\nabla_{\mathbf{a}_t} \log \hat{\pi}_t(\mathbf{a}_t|\mathbf{s}) = \nabla_{\mathbf{a}_t} \log \pi_{t,\theta}(\mathbf{a}_t|\mathbf{s}) + w\nabla_{\mathbf{a}_t} \log p_\phi(y|\mathbf{a_t}, \mathbf{s}, t) \tag{5}$$

Here, the variable $y$ in the classifier $p_\phi(y|\mathbf{a}_t, \mathbf{s}, t)$ is defined as a binary optimality variable, with $y \in \{0, 1\}$ and $y = 1$ indicating that the action $\mathbf{a}_t$ at $(\mathbf{s}, t)$ is optimal. We model this classifier in an energy-based form as $p_\phi(y = 1|\mathbf{a}_t, \mathbf{s}, t) \propto \exp(Q_\phi(\mathbf{s}, \mathbf{a}_t, t))$, where $Q_\phi(\mathbf{s}, \mathbf{a}_t, t)$ denotes a noise-level critic that conditions on both the noised action $\mathbf{a}_t$ and the diffusion timestep $t$. Under this definition, the gradient of the classifier's log-likelihood becomes

$$\nabla_{\mathbf{a}_t} \log p_\phi(y = 1|\mathbf{a}_t, \mathbf{s}, t) = \nabla_{\mathbf{a}_t} Q_\phi(\mathbf{s}, \mathbf{a}_t, t), \tag{6}$$

which shows that the gradient from the classifier aligns with the critic gradient. Thus, we can rewrite the Equation (5) as follows:

$$\nabla_{\mathbf{a}_t} \log \hat{\pi}_t(\mathbf{a}_t|\mathbf{s}) = \nabla_{\mathbf{a}_t} \log \pi_{t,\theta}(\mathbf{a}_t|\mathbf{s}) + w\nabla_{\mathbf{a}_t} Q_\phi(\mathbf{s}, \mathbf{a}_t, t) \tag{7}$$

This gradient representation corresponds to the policy of the form $\hat{\pi}_t(\mathbf{a}_t|\mathbf{s}) = \pi_{t,\theta}(\mathbf{a}_t|\mathbf{s}) \cdot \exp(wQ_\phi(\mathbf{s}, \mathbf{a}_t, t))/Z_t(\mathbf{s})$, with $Z_t(\mathbf{s}) = \int \pi_{t,\theta}(\mathbf{a}_t|\mathbf{s}) \cdot \exp(wQ_\phi(\mathbf{s}, \mathbf{a}_t, t))d\mathbf{a}_t$, which in turn arises as the solution of the KL-regularized optimization:

$$\hat{\pi}_t(\mathbf{a}_t|\mathbf{s}) = \arg\max_{\bar{\pi}} \mathbb{E}_{\mathbf{s}\sim\mathcal{B}, \mathbf{a}_t\sim\bar{\pi}(\cdot|\mathbf{s})} \left[ Q_\phi(\mathbf{s}, \mathbf{a}_t, t) - w^{-1} D_{KL}(\bar{\pi}(\cdot|\mathbf{s})\|\pi_{t,\theta}(\cdot|\mathbf{s})) \right] \tag{8}$$

This formulation maximizes the critic while constraining divergence from a reference policy $\pi_{t,\theta}$, in line with a behavior-regularized framework widely used in offline RL (Wu et al., 2019; Peng et al., 2019; Xu et al., 2023; Frans et al., 2025; Ki et al., 2025).

However, in online settings such a reference is unavailable or restrictive, making entropy maximization a natural alternative that encourages exploration. We therefore extend classifier guidance to the online RL setting by replacing the KL constraint in Equation (8) with an entropy term:

$$\hat{\pi}_t(\mathbf{a}_t|\mathbf{s}) = \arg\max_{\bar{\pi}} \mathbb{E}_{\mathbf{s}\sim\mathcal{B}, \mathbf{a}_t\sim\bar{\pi}(\cdot|\mathbf{s})} \left[ Q_\phi(\mathbf{s}, \mathbf{a}_t, t) + w^{-1}\mathcal{H}(\bar{\pi}(\cdot|\mathbf{s})) \right] \tag{9}$$

This represents the special case where $\pi_{t,\theta}$ in Equation (8) is uniform over actions, and the guided policy consequently simplifies to a Boltzmann distribution, closely resembling the soft policies widely adopted in online RL (Haarnoja et al., 2017; 2018a; Jain et al., 2024; Ma et al., 2025):

$$\hat{\pi}_t(\mathbf{a}_t|\mathbf{s}) = \exp(wQ_\phi(\mathbf{s}, \mathbf{a}_t, t))/Z_t(\mathbf{s}), \quad \text{where } Z_t(\mathbf{s}) = \int \exp(wQ_\phi(\mathbf{s}, \mathbf{a}_t, t))d\mathbf{a}_t. \tag{10}$$

Differentiating the logarithm of the policy $\hat{\pi}_t$ yields

$$\nabla_{\mathbf{a}_t} \log \hat{\pi}_t(\mathbf{a}_t|\mathbf{s}) = w\nabla_{\mathbf{a}_t} Q_\phi(\mathbf{s}, \mathbf{a}_t, t). \tag{11}$$

This formulation removes the dependence on the score network $\pi_{t,\theta}$ in Equation (7) by allowing the critic's gradient field to directly guide the denoising process. Based on this observation, the resulting denoising process can be expressed purely through the $Q$-function, as formalized below.

---

**Algorithm 1** Actor-Critic w/o Actor (ACA)

---

**Input:** Replay buffer $\mathcal{B}$, guidance weight $w$, noise-level critic $Q_\phi(\mathbf{s}, \mathbf{a}_t, t)$, denoising step $T$

1: **for** each iteration **do**
2:     **for** each sampling step **do**
3:         Sample $\mathbf{a}_0 \sim \pi_Q(\cdot|\mathbf{s})$ by Definition 1
4:         Execute $\mathbf{a}_0$, observe reward $r$ and next state $\mathbf{s}'$
5:         Store transition $(\mathbf{s}, \mathbf{a}_0, r, \mathbf{s}')$ in buffer $\mathcal{B}$
6:     **for** each update step **do**
7:         Sample minibatch from $\mathcal{B}$
8:         Update Critic $Q_\phi$ with Equation (12)

---

**Definition 1** (**Critic-guided denoising process**). *From Equation* (11)*, the score can be equivalently expressed in the form of a noise-prediction network*

$$\hat{\epsilon}(\mathbf{a}_t, \mathbf{s}, t) = -w\sigma_t \nabla_{\mathbf{a}_t} Q_\phi(\mathbf{s}, \mathbf{a}_t, t).$$

*Substituting this guided noise into the reverse diffusion dynamics in Equation* (2)*, the reverse process can be reformulated directly in terms of the noise-level Q-function as:*

$$\mathbf{a}_{t-1} = \frac{1}{\sqrt{\alpha_t}} \left( \mathbf{a}_t + \frac{\beta_t}{\sqrt{1 - \bar{\alpha}_t}} w\sigma_t \nabla_{\mathbf{a}_t} Q_\phi(\mathbf{s}, \mathbf{a}_t, t) \right) + \sigma_t \mathbf{z}, \quad \mathbf{z} \sim \mathcal{N}(\mathbf{0}, \mathbf{I}).$$

*Starting from $\mathbf{a}_T \sim \mathcal{N}(\mathbf{0}, \mathbf{I})$ and iterating to $t = 0$, we define the induced policy $\mathbf{a}_0 \sim \pi_Q(\cdot|\mathbf{s})$.*

Definition 1 defines a denoising process guided directly by the gradient of the noise-level critic, without requiring a separately trained noise-prediction network. The resulting policy $\pi_Q$ acts as an implicit actor that generates actions by iteratively refining Gaussian noise under the critic's gradient field. In contrast to conventional actor-critic methods, where the explicit actor typically lags behind the critic, $\pi_Q$ maintains immediate alignment between sampled actions and the critic's current value estimates. This mechanism enables $\pi_Q$ to encourage both high-value and high-entropy behavior, thereby achieving policy improvement without an explicit actor. Building on this formulation, we introduce **Actor-Critic without Actor (ACA)**, which eliminates the actor network entirely while retaining policy improvement through the critic-guided denoising process.

## 3.2 ACTOR-CRITIC WITHOUT ACTOR

Based on Definition 1, we formalize ACA as an actor-critic algorithm in which the actor role is entirely replaced by critic-guided denoising. The overall procedure is summarized in Algorithm 1, highlighting the simplicity of our algorithm.

**Critic objective**     The noise-level critic $Q_\phi(\mathbf{s}, \mathbf{a}_t, t)$ is trained with a two-part objective that anchors values at the denoised endpoint ($t = 0$) and propagates them to noisy timesteps ($t > 0$):

$$\min_\phi \; \mathbb{E}_{\mathbf{s}, \mathbf{a}_0, \mathbf{s}' \sim \mathcal{B}, \mathbf{a}_0' \sim \pi_Q(\cdot|\mathbf{s}')} \left[ \left( Q_\phi(\mathbf{s}, \mathbf{a}_0, 0) - \left( r(\mathbf{s}, \mathbf{a}_0) + \gamma Q_{\bar{\phi}}(\mathbf{s}', \mathbf{a}_0', 0) \right) \right)^2 \right] \tag{12}$$

$$+ \mathbb{E}_{\mathbf{s}, \mathbf{a}_0 \sim \mathcal{B}, t \sim \mathcal{U}[1,T], \epsilon \sim \mathcal{N}(\mathbf{0}, \mathbf{I})} \left[ \left( Q_\phi\left(\mathbf{s}, \sqrt{\bar{\alpha}_t}\mathbf{a}_0 + \sqrt{1 - \bar{\alpha}_t}\epsilon, t\right) - \texttt{stop\_grad}(Q_\phi(\mathbf{s}, \mathbf{a}_0, 0)) \right)^2 \right],$$

where $Q_{\bar{\phi}}$ is a target network and $\mathcal{U}[1, T]$ denotes the uniform distribution over timesteps. The first loss term corresponds to a standard temporal difference (TD) regression, with the only difference being that the next action $\mathbf{a}_0'$ is sampled by the implicit actor $\pi_Q$ rather than the explicit parameterized policy. The second loss term regresses $Q_\phi(\mathbf{s}, \mathbf{a}_t, t)$ toward the fully denoised value $Q_\phi(\mathbf{s}, \mathbf{a}_0, 0)$ with gradients stopped at the target, effectively transporting value information across the denoising chain.

**Eliminating policy optimization**     A central contribution of ACA is the complete removal of the explicit actor network. Conventional actor-critic frameworks must separately train an actor to track the critic, which introduces additional optimization complexity, hyperparameter sensitivity, and inevitable policy lag as the actor cannot instantly reflect the critic's most recent updates. ACA

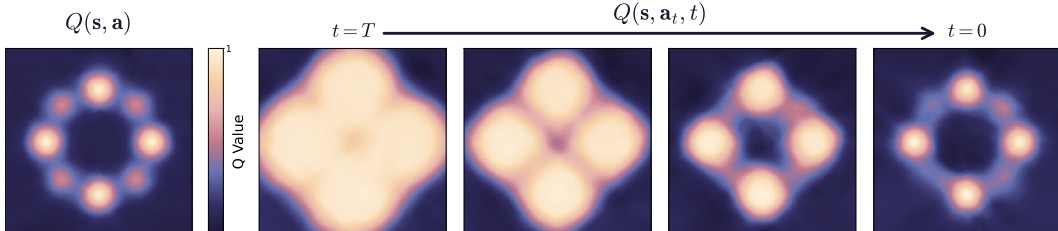

Figure 2: Visualization of value maps in 2D bandit environment for noise-level $Q(\mathbf{s}, \mathbf{a}_t, t)$ across different diffusion steps ($t = 9, 6, 3, 0$) compared to the standard Bellman critic $Q(\mathbf{s}, \mathbf{a})$. Detailed setup in the 2D bandit environment is provided in Appendix A.1.

circumvents these challenges by discarding the actor and directly generating actions through the gradient field of a noise-level critic. This design ensures that behavior remains immediately aligned with up-to-date value estimates, tightly coupling evaluation and improvement without the overhead of actor optimization. As a result, ACA achieves a lightweight architecture that avoids the difficulties of actor learning while consistently maintaining alignment between the critic and behavior.

**Noise-level critic $Q_t$**    A crucial component of ACA is the noise-level critic $Q(\mathbf{s}, \mathbf{a}_t, t)$, which conditions on both the noised action $\mathbf{a}_t$ and the diffusion timestep $t$. Unlike a standard Bellman critic that only evaluates terminal actions, $Q_t$ provides value estimates throughout the denoising process, ensuring that guidance remains informative even under substantial noise corruption.

**Proposition 1 (Noise-level critic consistency).** *For any fixed $\mathbf{s}$, the population minimizer of the noisy timestep loss ($t > 0$) satisfies*

$$Q(\mathbf{s}, \mathbf{a}_t, t) = \mathbb{E}_{\mathbf{a}_0 \sim q(\mathbf{a}_0 | \mathbf{a}_t, \mathbf{s}, t)} \left[ Q(\mathbf{s}, \mathbf{a}_0, 0) \right].$$

Proposition 1 formalizes that $Q_t$ approximates the conditional expectation of the terminal value $Q(\mathbf{s}, \mathbf{a}_0, 0)$ with respect to the posterior defined by the forward diffusion process. This regression consistency induces a smoothing effect across noise levels, ensuring that gradients remain stable even when $\mathbf{a}_t$ lies in highly corrupted regions. Consequently, the gradient field $\nabla_{\mathbf{a}_t} Q(\mathbf{s}, \mathbf{a}_t, t)$ provides reliable guidance for denoising, enabling actions to converge toward globally consistent high-value modes. As illustrated in Figure 2, this smoothing property allows value information to generalize coherently across the entire diffusion chain, in contrast to the standard Bellman critic $Q(\mathbf{s}, \mathbf{a})$ that lacks such noise-aware regularization.

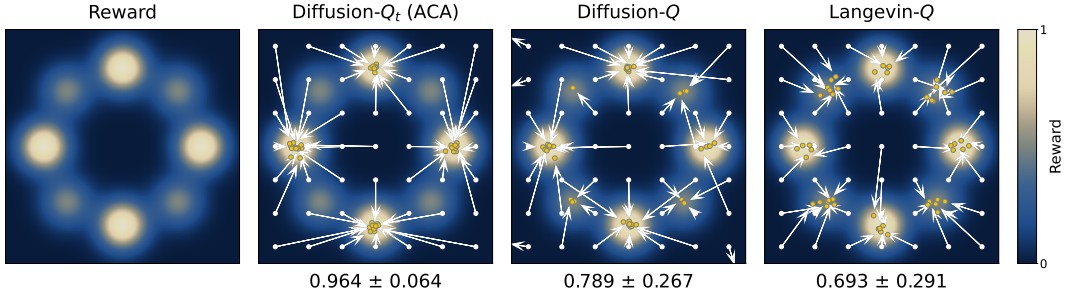

Figure 3: Visualization of sampled actions $\mathbf{a}_0$ obtained from different reverse processes in the 2D bandit environment. The leftmost panel shows the reward landscape of the environment. White dots denote initial samples $\mathbf{a}_T$, and arrows indicate their corresponding denoised actions $\mathbf{a}_0$ (yellow) guided by each method. The numbers below each plot show the mean reward $\pm$ standard deviation, computed over 10k denoised samples starting from $\mathbf{a}_T \sim \mathcal{N}(\mathbf{0}, \mathbf{I})$. The detailed setup of the 2D bandit environment is provided in Appendix A.1.

**Illustrative examples**    The advantage of employing the noise-level critic is further illustrated in Figure 3, which visualizes sampled actions from different reverse processes in the 2D bandit environment. The figure compares three approaches: **Diffusion-$Q_t$ (ACA)**, which performs denoising

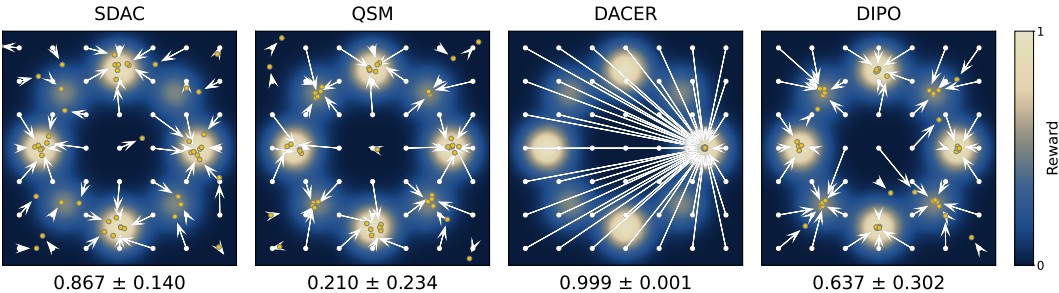

Figure 4: Visualization of sampled actions $\mathbf{a}_0$ obtained from baseline algorithms (SDAC, QSM, DACER, and DIPO) under the same 2D bandit setting as in Figure 3.

guided by the noise-level critic; **Diffusion-$Q$**, which substitutes $Q_t$ with a standard Bellman critic $Q$; and **Langevin-$Q$**, which replaces the diffusion denoising process with Langevin dynamics. This comparison isolates two factors: (i) the critic parameterization ($Q_t$ vs. $Q$), and (ii) the choice of reverse process. Diffusion denoising unfolds over multiple timesteps, where the variance schedule gradually reduces noise, providing a multi-scale refinement of actions from coarse to fine resolutions. In contrast, Langevin updates proceed at a single scale, applying the critic's gradient directly at each step. The detailed algorithm for **Langevin-$Q$** is provided in Appendix C.

As illustrated in Figure 3, Diffusion-$Q_t$ (ACA) enables coarse-to-fine refinement of actions, providing stable guidance throughout denoising and effectively steering samples toward distinct high-value modes. This is achieved by conditioning value estimates on noise levels, with gradients adaptively scaled by the variance schedule. In contrast, Diffusion-$Q$, which lacks timestep conditioning, often produces brittle gradients under high noise, making it prone to spurious local minima. Langevin-$Q$ applies gradients at a fixed scale, foregoing progressive rescaling and thus spreading samples broadly across the action space without consistently capturing high-value modes. Overall, these comparisons demonstrate that combining diffusion denoising with a noise-level critic yields well-conditioned gradients and reliable coverage of high-value actions.

**Multi-modal action coverage**   Multi-modality is a critical property of RL policies, as it allows the representation of diverse high-value behaviors rather than collapsing into a single deterministic solution (Haarnoja et al., 2017). Preserving multiple modes facilitates exploration of complex reward landscapes, maintains behavioral diversity, and improves robustness to downstream tasks. Figure 3 demonstrates that ACA successfully captures all four high-value modes in the 2D bandit environment by generating diverse actions through critic-guided denoising. In contrast, diffusion-based baselines in Figure 4 collapse into a single dominant mode or yield uneven sample distributions.

Beyond qualitative comparisons in Figure 3 and 4, Table 1 provides a quantitative evaluation of multi-modality in the 2D bandit environment. The table reports the proportion of samples reaching each of the four high-value modes, measured over 10k samples denoised from $\mathbf{a}_T \sim \mathcal{N}(\mathbf{0}, \mathbf{I})$. DACER (Wang et al., 2024) collapses entirely into a single mode because it directly trains the diffusion model to maximize $Q$, which drives samples toward the highest-valued region and induces severe mode collapse. By contrast, QSM (Psenka et al., 2023) and DIPO (Yang et al., 2023) do not leverage a noise-level critic $Q_t$, leading

Table 1: Proportion of samples reaching each high-value mode. Detailed explanation is provided in Appendix A.2.

| Method | Proportions | | | | Sum |
|---|---|---|---|---|---|
| SDAC | 0.227 | 0.227 | 0.239 | 0.234 | 0.927 |
| QSM | 0.118 | 0.115 | 0.117 | 0.115 | 0.465 |
| DACER | 0.000 | 1.000 | 0.000 | 0.000 | 1.000 |
| DIPO | 0.100 | 0.099 | 0.104 | 0.098 | 0.401 |
| Langevin-$Q$ | 0.141 | 0.147 | 0.139 | 0.143 | 0.570 |
| Diffusion-$Q$ | 0.141 | 0.182 | 0.160 | 0.153 | 0.636 |
| **ACA (Ours)** | 0.240 | 0.256 | 0.243 | 0.254 | 0.993 |

to misaligned gradients during denoising and consequently insufficient coverage of high-value modes (0.465 and 0.401). SDAC (Ma et al., 2025), on the other hand, preserves multi-modality more effectively by employing a carefully devised diffusion training objective, achieving a score of 0.927. Nonetheless, this comes at the cost of algorithmic complexity, as SDAC requires multiple auxiliary tricks and incurs substantial computational overhead due to its diffusion actor. Unlike the baselines, ACA employs a critic-guided denoising process in place of an explicit actor network, thereby avoiding high architectural complexity and achieving an aggregate score of 0.993 with nearly uniform sample proportions across all four modes ($\approx$0.25).

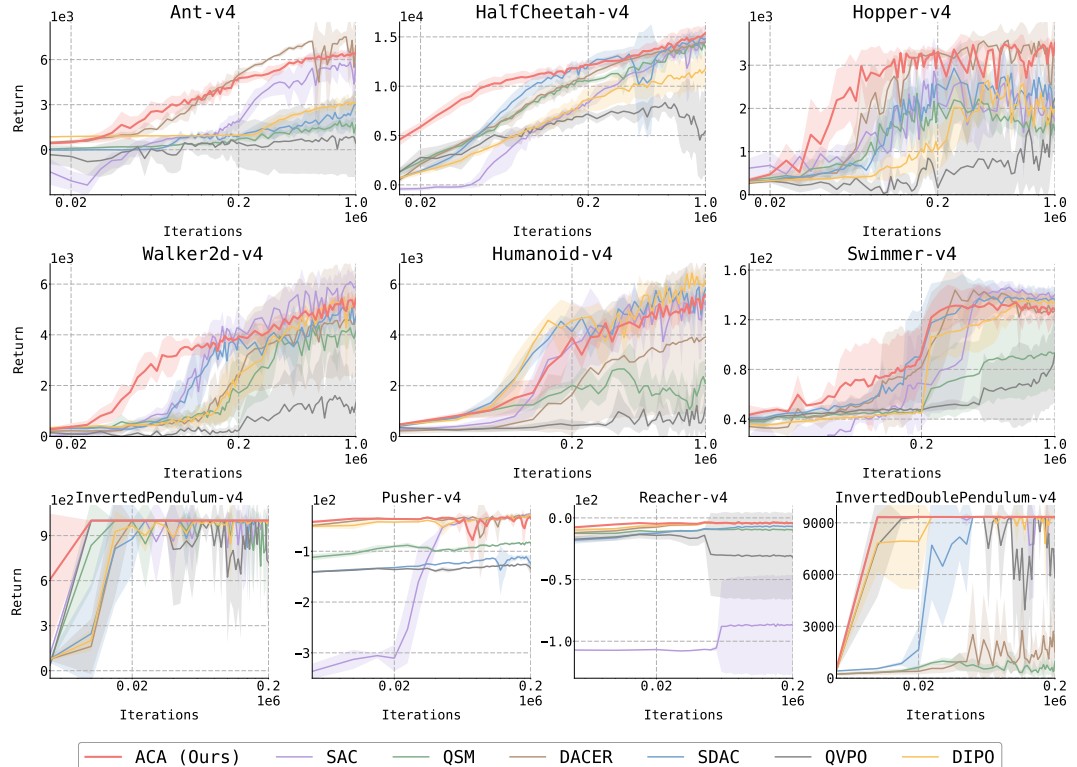

Figure 5: Training performance on OpenAI Gym MuJoCo environments. Each curve reports the mean return over 5 random seeds, with shaded regions denoting the 95% confidence interval.

**Summary** Standard actor-critic methods incur substantial algorithmic overhead, which is further amplified when diffusion models are used as actors due to their large networks and additional design complexity. ACA circumvents such burdens by eliminating the explicit actor network and instead generating actions directly from the gradient field of a noise-level critic, ensuring immediate alignment between actions and value estimates. The noise-level critic further stabilizes training by propagating terminal values across noise levels, yielding well-conditioned gradients even under severe corruption. Moreover, ACA faithfully preserves the multi-modal structure of action distributions, enabling balanced coverage of diverse high-value behaviors and robust exploration.

## 4 EXPERIMENTS

### 4.1 ONLINE RL

We evaluate the online RL performance of ACA on a suite of MuJoCo control tasks from OpenAI Gym. As baselines, we consider the standard off-policy actor-critic algorithm SAC (Haarnoja et al., 2018a) along with several diffusion-based actor-critic methods: QSM (Psenka et al., 2023), DIPO (Yang et al., 2023), QVPO (Ding et al., 2024), DACER (Wang et al., 2024), and SDAC (Ma et al., 2025). Additional experimental details are provided in Appendix D.1.

Figure 5 shows training curves across 10 tasks. Algorithms are trained for 1M steps on six environments (Ant-v4, HalfCheetah-v4, Hopper-v4, Walker2d-v4, Humanoid-v4, and Swimmer-v4) and for 200k steps on four environments (Pusher-v4, Reacher-v4, InvertedPendulum-v4, and InvertedDoublePendulum-v4). ACA achieves faster performance gains with fewer interactions than baselines, while attaining competitive or superior final returns. Table 2 further reports performance at 100k steps, highlighting ACA's advantage in early-stage learning. Although ACA exhibits slower improvement than DIPO and SDAC on Humanoid-v4, it remains more parameter-efficient and achieves stronger results across the other environments. These improvements stem from ACA's actor-free design, which eliminates policy lag and aligns sampled

Table 2: Performance at 100k steps, reported as mean return $\pm$ 95% confidence interval over 5 seeds.

| | w/ Actor | | | | | | w/o Actor |
| | SAC | QSM | DIPO | DACER | QVPO | SDAC | ACA |
|---|---|---|---|---|---|---|---|
| Ant-v4 | $884 \pm 44$ | $397 \pm 36$ | $932 \pm 33$ | $2623 \pm 758$ | $380 \pm 363$ | $811 \pm 113$ | $\mathbf{3044 \pm 504}$ |
| HalfCheetah-v4 | $5691 \pm 659$ | $8389 \pm 614$ | $5831 \pm 782$ | $8990 \pm 696$ | $5622 \pm 943$ | $10364 \pm 835$ | $\mathbf{11206 \pm 575}$ |
| Hopper-v4 | $962 \pm 1056$ | $1366 \pm 428$ | $664 \pm 354$ | $2420 \pm 740$ | $61 \pm 103$ | $1538 \pm 665$ | $\mathbf{2960 \pm 312}$ |
| Walker2d-v4 | $2262 \pm 611$ | $755 \pm 283$ | $776 \pm 363$ | $621 \pm 319$ | $325 \pm 190$ | $1816 \pm 898$ | $\mathbf{3510 \pm 332}$ |
| Humanoid-v4 | $789 \pm 317$ | $1226 \pm 298$ | $2217 \pm 955$ | $522 \pm 326$ | $321 \pm 78$ | $\mathbf{2274 \pm 420}$ | $1513 \pm 665$ |
| Swimmer-v4 | $42.0 \pm 5.2$ | $45.3 \pm 1.1$ | $42.2 \pm 0.7$ | $53.0 \pm 6.7$ | $47.3 \pm 0.8$ | $53.7 \pm 5.4$ | $\mathbf{72.0 \pm 29.2}$ |

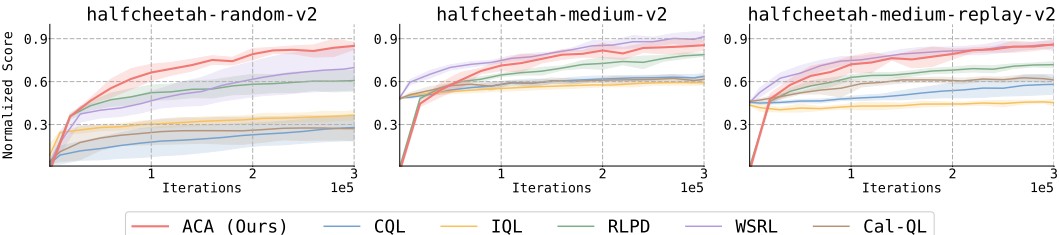

Figure 6: Training performance on `HalfCheetah-v2` environment with each suboptimal offline dataset. Each curve reports the mean return over 5 random seeds with 95% confidence interval. Results are shown for the online training phase, while offline pre-training is omitted.

actions immediately with critic updates, as well as from its critic-guided denoising mechanism, which preserves multi-modality and supports a balanced exploration–exploitation trade-off. Overall, ACA is both sample-efficient and capable of attaining favorable learning curves than competing baselines.

Beyond performance, we also evaluate the model complexity of ACA relative to baseline methods. Table 3 reports parameter counts in the `Humanoid-v4` environment, where ACA requires substantially fewer parameters as a result of removing the explicit actor network. While ACA uses only 475k parameters (0.677), which is substantially smaller than diffusion-based algorithms such as QSM, DIPO, DACER, QVPO, and SDAC, and even smaller than SAC with 702k parameters (normalized to 1.0). This lightweight design reduces architectural and hyperparameter complexity while maintaining competitive performance, establishing ACA as a practical alternative to actor-based methods.

Table 3: Normalized parameter counts.

| Method | # Params |
|---|---|
| SAC | 1.000 |
| QSM | 1.000 |
| DIPO | 1.012 |
| DACER | 1.008 |
| QVPO | 1.007 |
| SDAC | 1.007 |
| **ACA (Ours)** | **0.677** |

## 4.2 ONLINE RL WITH OFFLINE DATASETS

We further evaluate ACA against offline-to-online baselines to assess whether it achieves more favorable learning curves while maintaining efficiency in settings where sample efficiency is particularly critical. The baselines include the offline RL algorithms CQL (Kumar et al., 2020) and IQL (Kostrikov et al., 2021), the offline-to-online algorithms Cal-QL (Nakamoto et al., 2023) and WSRL (Zhou et al., 2024), and the efficient online RL algorithm RLPD (Ball et al., 2023), which learns entirely from scratch by constructing each mini-batch as an equal mixture (50/50) of samples from the offline dataset and the online replay buffer. In this setup, ACA adopts the same protocol as RLPD, starting directly from online learning without offline pre-training. By contrast, CQL, IQL, Cal-QL, and WSRL are trained for 250k offline steps before transitioning to the online phase. Moreover, whereas all baselines employ ensembles of ten $Q$-networks, ACA relies only on a standard double-$Q$ setup. As shown in Figure 6, ACA consistently matches or outperforms these algorithms across diverse suboptimal dataset conditions, while maintaining efficiency by avoiding large ensembles and operating without any offline pre-training. Detailed experimental settings are provided in Appendix D.2.

## 4.3 ABLATION STUDIES

We conduct ablation studies to examine the effect of the guidance weight $w$ and the number of denoising steps $T$ on ACA's performance. As shown in Figure 7, we sweep $w \in \{1, 5, 10, 30, 50, 100\}$ and $T \in \{5, 10, 20, 50, 100\}$ while keeping other hyperparameters fixed. The guidance weight controls

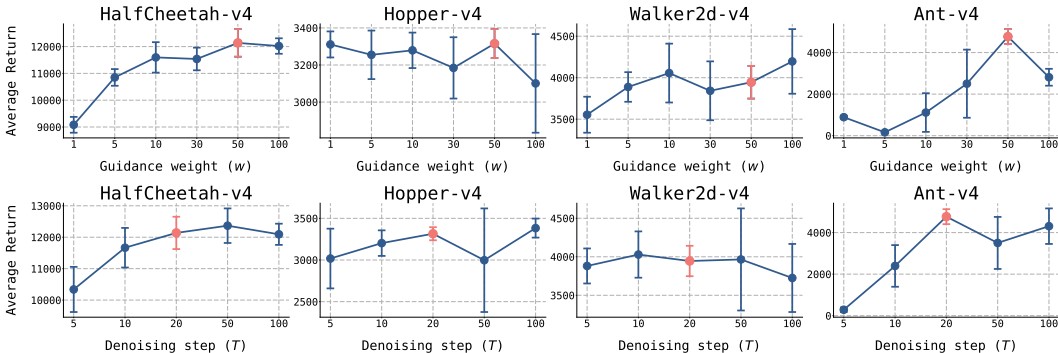

Figure 7: Performance of ACA across MuJoCo environments under varying guidance weight $w$ and denoising steps $T$, evaluated at 200k steps. The default hyperparameters are highlighted in red.

the balance between $Q$-maximization and entropy maximization: small values ($w = 1, 5$) emphasize entropy and induce overly exploratory behavior, whereas large values ($w = 100$) suppress exploration and yield greedy actions. Intermediate settings ($w = 30, 50$) provide the best trade-off. For denoising steps, small values ($T = 5, 10$) result in poor performance, while larger values yield comparable returns. A setting of $T = 20$ offers strong performance with higher efficiency, making it the most practical choice.

## 5 RELATED WORKS

**Diffusion models in offline RL**   Diffusion models have recently been established as powerful policy representations in offline RL, providing a natural way to capture multi-modal behaviors. Wang et al. (2022) introduce conditional diffusion models that combine behavior cloning with $Q$-learning to achieve strong performance. Janner et al. (2022) propose trajectory-level denoising for planning, enabling long-horizon reasoning and flexible goal conditioning. Chen et al. (2023) present a behavior-regularized policy optimization framework based on a pretrained diffusion behavior model, and Lu et al. (2023) formulate energy-guided sampling to realize principled $Q$-guided optimization.

**Diffusion models in online RL**   In online RL, diffusion policies have been adapted to support continual interaction and efficient policy improvement. Yang et al. (2023) establish the first formulation of diffusion policies with convergence guarantees. Ding et al. (2024) propose a variational lower bound on the policy objective, enabling sample-efficient online updates with entropy regularization. Wang et al. (2024) treat the reverse process itself as the policy, introducing adaptive exploration control through entropy estimation. Most recently, Ma et al. (2025) generalize denoising objectives to train policies directly on value-based targets, yielding efficient online algorithms.

## 6 CONCLUSION AND LIMITATIONS

In this work, we introduce **Actor-Critic without Actor (ACA)**, a lightweight framework that eliminates the explicit actor network and replaces standard policy improvement with critic-guided denoising. Through extensive experiments, we show that ACA achieves more favorable learning curves and shows competitive or superior performance compared to both standard actor-critic methods and diffusion-based approaches, while requiring fewer parameters and simpler training.

**Limitations**   Despite these advantages, ACA requires sampling actions through an iterative denoising process when training the critic with the Bellman operator, which is computationally more expensive than algorithms such as SAC or PPO (Schulman et al., 2017) that do not rely on iterative denoising. Moreover, ACA currently lacks an automatic mechanism for adjusting the guidance weight $w$, which must be tuned manually, similar to entropy regularization in other RL algorithms (Schulman et al., 2017; Psenka et al., 2023; Ding et al., 2024). Future work includes extending ACA with soft $Q$-functions (Haarnoja et al., 2017; 2018a) to better capture entropy-regularized objectives and developing adaptive strategies for automatic guidance-weight tuning.

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

## A IMPLEMENTATION DETAILS IN 2D BANDIT ENVIRONMENTS

### A.1 ENVIRONMENT SETTINGS

We design a multi-modal reward function based on a mixture of Gaussian distributions, as illustrated in Figure 8. The reward corresponds to the probability density of this mixture, resulting in a landscape with eight modes, each represented by an isotropic Gaussian with covariance $0.3^2\mathbf{I}$. To induce asymmetry, alternating weights of 2 and 1 are assigned to the modes, which are positioned on a circle of radius $\sqrt{2}$ at coordinates $[(\sqrt{2}, 0), (1, 1), (0, \sqrt{2}), (-1, 1), (-\sqrt{2}, 0), (-1, -1), (0, -\sqrt{2}), (1, -1)]$. This arrangement produces alternating high- and low-reward regions around the circle. The reward values are normalized so that the maximum equals 1.0. This structure highlights how ACA's smooth value function helps avoid convergence to local optima by effectively navigating multiple reward modes across the state space.

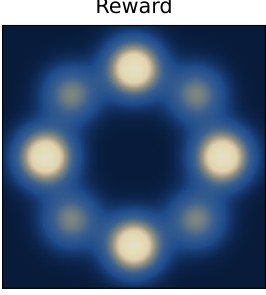

Reward

Figure 8: Reward map.

**Training details** We select the guidance weight $w$ for Diffusion-$Q_t$ (ACA), Diffusion-$Q$, and Langevin-$Q$ by measuring the average reward over 10k samples for $w \in [1, 400]$. The optimal values obtained from this sweep are used in the evaluations and action sampling shown in Figure 3.

### A.2 EVALUATION OF MULTI-MODALITY

In Table 1, we report the proportion of samples assigned to each high-value mode for all methods. Sampling starts by drawing $\mathbf{a}_T \sim \mathcal{N}(\mathbf{0}, \mathbf{I})$ and propagating through each algorithm's sampling process. The four proportions correspond, in order from left to right, to the top, right, bottom, and left high-value modes. Each proportion is computed as the ratio of samples lying within an $\mathcal{L}_2$-distance of 0.3 from the mode center to the total number of samples (10k).

## B FULL VISUALIZATIONS ON 2D BANDIT ENVIRONMENT

To reveal the intermediate denoising samples not shown in Figure 3 and Figure 4, we provide visualizations at each denoising step. In this setting, initial actions $\mathbf{a}_T$ are sampled from a grid rather than the standard normal distribution, and we fix the number of denoising steps to $T = 10$ for all baselines.

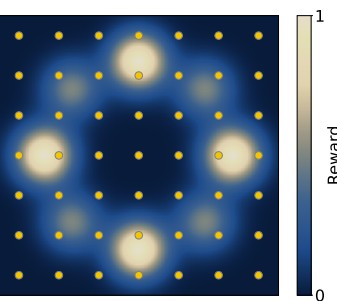

Figure 9: Initial samples $\mathbf{a}_T$.

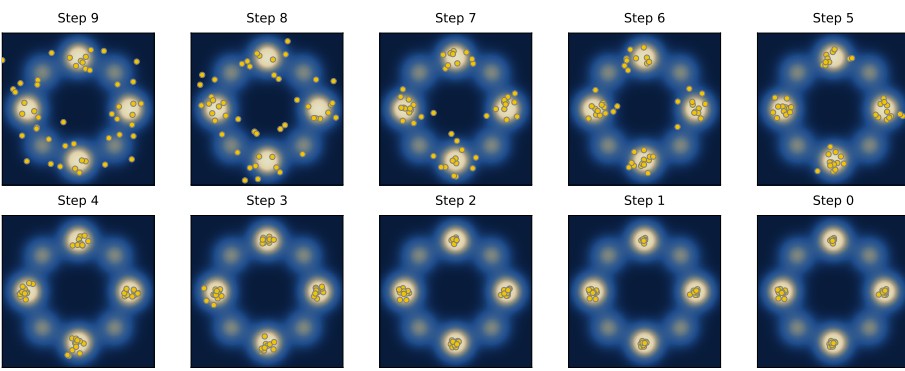

Figure 10: Visualizations of our method (ACA).

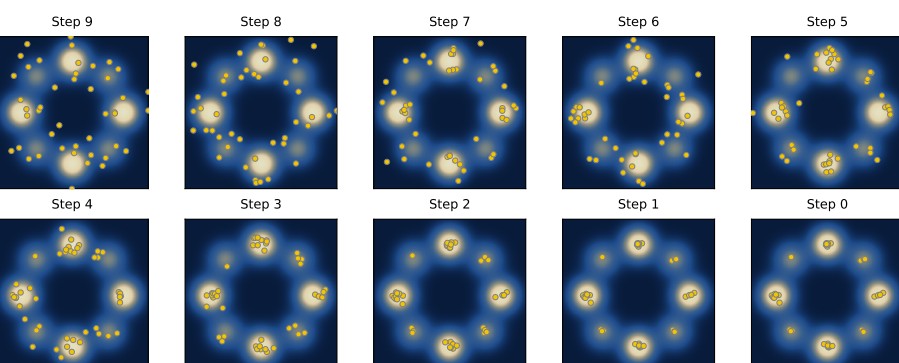

Figure 11: Visualizations of Diffusion-$Q$.

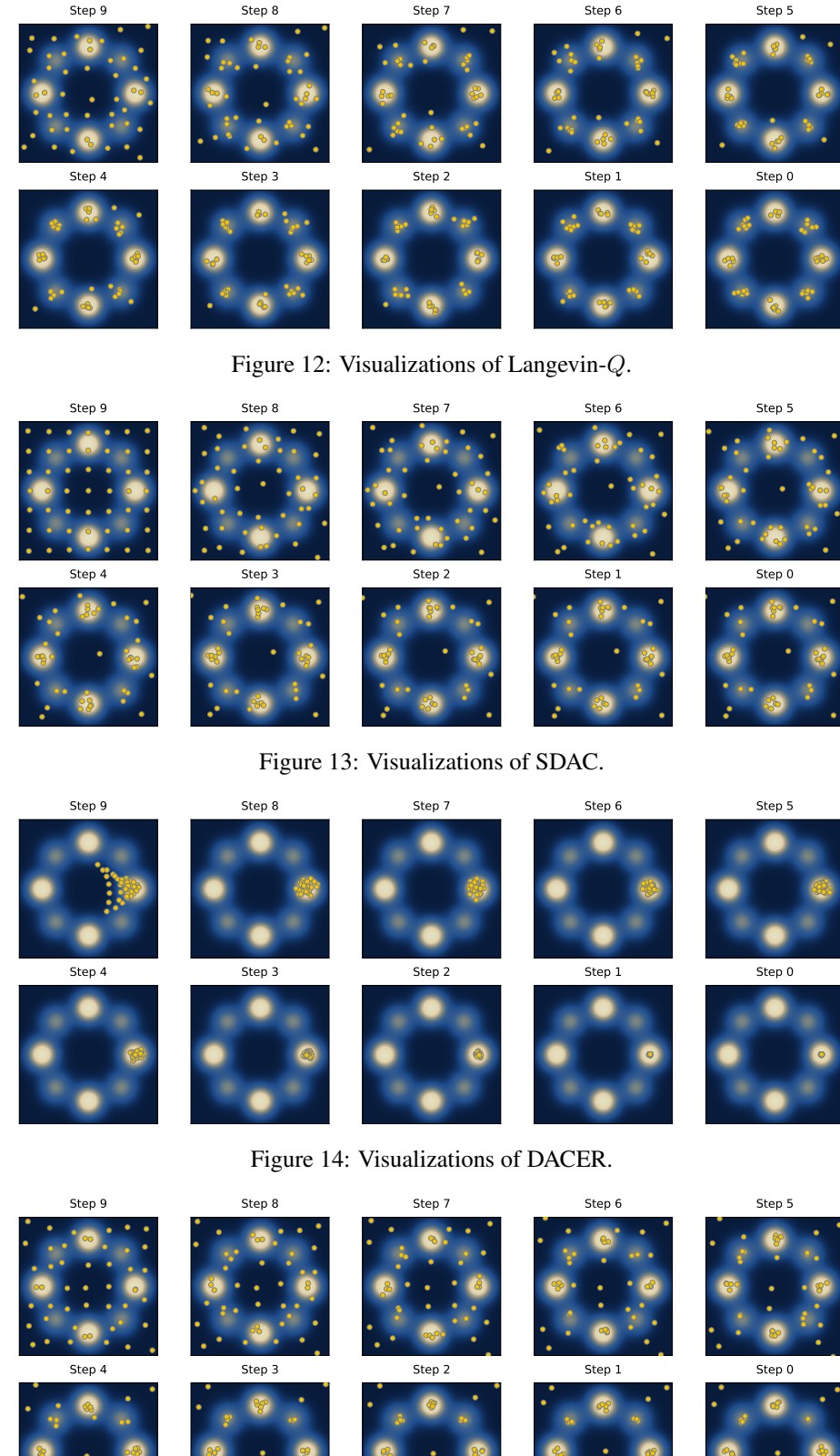

Figure 12: Visualizations of Langevin-$Q$.

Figure 13: Visualizations of SDAC.

Figure 14: Visualizations of DACER.

Figure 15: Visualizations of QSM.

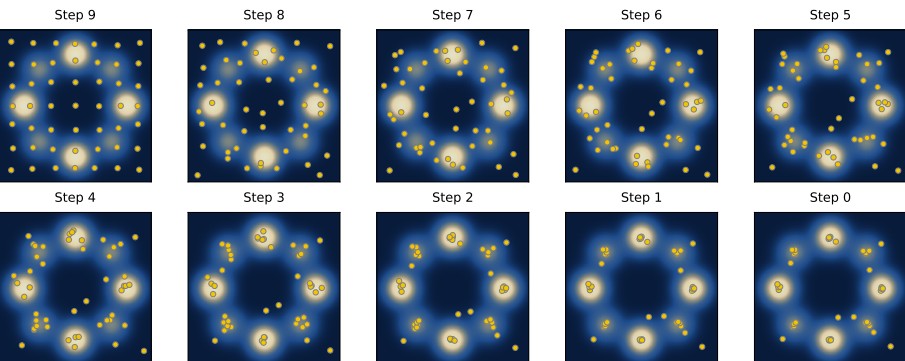

Figure 16: Visualizations of DIPO.

## C  CRITIC-GUIDED LANGEVIN DYNAMICS

---
**Algorithm 2** Langevin-$Q$
---
**Input:** Replay buffer $\mathcal{B}$, step size $\epsilon$, guidance weight $w$, critic $Q_\phi(\mathbf{s}, \mathbf{a})$, denoising step $T$
1: **for** each iteration **do**
2:     **for** each sampling step **do**
3:         Sample $\mathbf{a}_0 \sim \pi_L(\cdot|\mathbf{s})$ by Definition 2
4:         Execute $\mathbf{a}_0$, observe reward $r$ and next state $\mathbf{s}'$
5:         Store transition $(\mathbf{s}, \mathbf{a}_0, r, \mathbf{s}')$ in buffer $\mathcal{B}$
6:     **for** each update step **do**
7:         Sample mini-batch from $\mathcal{B}$
8:         Update Critic $Q_\phi$ with $\mathbb{E}_{\mathbf{s}, \mathbf{a}_0, \mathbf{s}' \sim \mathcal{B}, \mathbf{a}_0' \sim \pi_L(\cdot|\mathbf{s}')} \left[ \left( Q_\phi(\mathbf{s}, \mathbf{a}_0) - \left( r(\mathbf{s}, \mathbf{a}) + \gamma Q_{\bar{\phi}}(\mathbf{s}', \mathbf{a}_0') \right) \right)^2 \right]$
---

Instead of relying on the diffusion models' denoising process, one can sample from the Boltzmann policy $\pi(\mathbf{a}|\mathbf{s}) = \exp\left(wQ(\mathbf{s}, \mathbf{a})\right)/Z(\mathbf{s})$, where $Z(\mathbf{s}) = \int \exp(wQ(\mathbf{s}, \mathbf{a}))d\mathbf{a}$, using Langevin dynamics. Langevin dynamics generates samples from a target distribution $p(\mathbf{x})$ given access to its score $\nabla_\mathbf{x} \log p(\mathbf{x})$. With a fixed step size $\epsilon > 0$, the reverse process is defined as:

$$\mathbf{x}_{t-1} = \mathbf{x}_t + \frac{\epsilon}{2}\nabla_\mathbf{x} \log p(\mathbf{x}_t) + \sqrt{\epsilon}\mathbf{z}_t,$$

where $\mathbf{z}_t \sim \mathcal{N}(\mathbf{0}, \mathbf{I})$. As $\epsilon \to 0$ and $T \to \infty$, the distribution of $\mathbf{x}_T$ converges to $p(\mathbf{x})$ under mild regularity conditions (Welling & Teh, 2011). In practice, approximate samples can be obtained with finite $T$ and sufficiently small $\epsilon$. Applying this principle to the Boltzmann policy $\pi(\mathbf{a}|\mathbf{s}) \propto \exp(wQ(\mathbf{s}, \mathbf{a}))$, we obtain the following sampling process:

**Definition 2** (Critic-guided Langevin dynamics). *Starting from Gaussian noise $\mathbf{a}_T \sim \mathcal{N}(\mathbf{0}, \mathbf{I})$ and applying the reverse Langevin update*

$$\mathbf{a}_{t-1} = \mathbf{a}_t + \frac{\epsilon}{2}\, w\nabla_\mathbf{a} Q(\mathbf{s}, \mathbf{a}_t) + \sqrt{\epsilon}\, \mathbf{z}_t, \quad \mathbf{z}_t \sim \mathcal{N}(\mathbf{0}, \mathbf{I}),$$

*sequentially for $t = T \to 1$, the resulting action is distributed as*

$$\mathbf{a}_0 \sim \pi_L(\cdot|\mathbf{s}),$$

*where $\pi_L$ denotes the implicit policy induced by the Langevin sampling procedure.*

Unlike the diffusion-based reverse process in Definition 1, this approach requires only a standard critic $Q_\phi(\mathbf{s}, \mathbf{a})$ trained via the Bellman operator, without introducing a noise-level critic $Q_\phi(\mathbf{s}, \mathbf{a}_t, t)$.

# D EXPERIMENTAL DETAILS

## D.1 ONLINE RL

Following Ma et al. (2025), we employed vectorized environments across five tasks. Consequently, the 1M training iterations reported in Figure 5 correspond to a total of 5M environment interactions. The hyperparameter configurations for the baseline algorithms are provided in Table 4, while those for ACA are summarized in Table 5. For the Humanoid-v4 environment, we set the target entropy to $-0.5 \cdot \dim(\mathcal{A})$ and the guidance weight to $w = 60.0$.

Table 4: Baseline algorithms' hyperparameter settings.

| Hyperparameter | SDAC | QSM | DIPO | DACER | QVPO | SAC |
|---|---|---|---|---|---|---|
| Replay buffer capacity | 1e6 | 1e6 | 1e6 | 1e6 | 1e6 | 1e6 |
| Buffer warm-up size | 3e4 | 3e4 | 3e4 | 3e4 | 3e4 | 3e4 |
| Batch size | 256 | 256 | 256 | 256 | 256 | 256 |
| Discount factor $\gamma$ | 0.99 | 0.99 | 0.99 | 0.99 | 0.99 | 0.99 |
| Target update rate $\tau$ | 0.005 | 0.005 | 0.005 | 0.005 | 0.005 | 0.005 |
| Reward scale | 0.2 | 0.2 | 0.2 | 0.2 | 0.2 | 0.2 |
| No. of hidden layers | 3 | 3 | 3 | 3 | 3 | 3 |
| No. of hidden nodes | 256 | 256 | 256 | 256 | 256 | 256 |
| Activations | Mish | ReLU | Mish | Mish | Mish | GELU |
| Diffusion steps | 20 | 20 | 100 | 20 | 20 | N/A |
| Action gradient steps | N/A | N/A | 30 | N/A | N/A | N/A |
| No. of Gaussian distributions | N/A | N/A | N/A | 3 | N/A | N/A |
| No. of action samples | N/A | N/A | N/A | 200 | N/A | N/A |
| Noise scale | 0.1 | N/A | N/A | 0.1 | N/A | N/A |
| Optimizer | Adam | Adam | Adam | Adam | Adam | Adam |
| Actor learning rate | 3e-4 | 3e-4 | 3e-4 | 3e-4 | 3e-4 | 3e-4 |
| Critic learning rate | 3e-4 | 3e-4 | 3e-4 | 3e-4 | 3e-4 | 3e-4 |
| Alpha learning rate | 7e-3 | N/A | N/A | 3e-2 | N/A | 3e-4 |
| Target entropy | $-0.9 \cdot \dim(\mathcal{A})$ | N/A | N/A | $-0.9 \cdot \dim(\mathcal{A})$ | N/A | $- \dim(\mathcal{A})$ |
| No. of batch action sampling | 32 | 32 | N/A | N/A | 32 | N/A |

Table 5: ACA's hyperparameter settings.

| Hyperparameter | ACA |
|---|---|
| Replay buffer capacity | 1e6 |
| Buffer warm-up size | 3e4 |
| Batch size | 256 |
| Discount $\gamma$ | 0.99 |
| Target network soft-update rate $\rho$ | 0.005 |
| Reward scale | 0.2 |
| No. of hidden layers | 3 |
| No. of hidden nodes | 256 |
| Activations in critic network | Mish |
| Diffusion steps | 20 |
| Critic delay update | 2 |
| Optimizer | Adam |
| Guidance weight | 50 |
| Critic learning rate | 1e-3 |
| No. of batch action sampling | 32 |
| Alpha learning rate | 3e-2 |
| Target entropy | $-0.9 \cdot \dim(\mathcal{A})$ |
| Noise scale | 0.1 |

## D.2 ONLINE RL WITH OFFLINE DATASETS

Following Zhou et al. (2024), the WSRL experiments in Figure 6 employ pre-trained policies and value functions obtained through CQL-based offline training rather than Cal-QL. This choice is motivated by the nature of the offline datasets, which contain dense rewards and lack terminal states, thereby precluding the availability of ground-truth return-to-go values required for the Cal-QL regularizer.

## E   PRACTICAL IMPLEMENTATIONS

**Batch action sampling**   For each state $\mathbf{s}$, we generate $N$ candidate actions by sampling Gaussian noise vectors $\mathbf{a}_T \sim \mathcal{N}(\mathbf{0}, \mathbf{I})$ and applying the denoising process. From these candidates, we select the action $\mathbf{a}_0$ that maximizes the terminal value $Q(\mathbf{s}, \mathbf{a}_0, 0)$. This sampling-selection strategy, also employed in prior diffusion-based RL methods (Ding et al., 2024; Ma et al., 2025), mitigates the stochasticity of the denoising process and facilitates more reliable exploitation. Furthermore, we add a Gaussian noise with an adaptively tuned noise level, following the approaches of Wang et al. (2022); Ma et al. (2025).

$Q$**-gradient normalization**   To further stabilize training, we normalize the critic gradient during denoising updates:

$$\nabla_{\mathbf{a}_t} Q_\phi(\mathbf{s}, \mathbf{a}_t, t) \leftarrow \nabla_{\mathbf{a}_t} Q_\phi(\mathbf{s}, \mathbf{a}_t, t) \big/ \left( \|\nabla_{\mathbf{a}_t} Q_\phi(\mathbf{s}, \mathbf{a}_t, t)\| + \epsilon \right).$$

This normalization prevents excessively large or uneven gradient magnitudes, which could otherwise lead to unstable updates. By ensuring a consistent scale, the denoising dynamics remain stable and the critic can learn smoother value estimates.

## F   USE OF LARGE LANGUAGE MODELS

In this work, large language models (LLMs) were employed in a limited capacity to assist with grammar correction, sentence refinement, and to improve the overall readability of the paper.

