# OpenReview forum: "Actor-Critic without Actor"
_ICLR.cc/2026/Conference — ICLR 2026 Conference Withdrawn Submission_

### Official Review · Reviewer_EhSP · 2025-10-27

**Soundness:** 1
**Presentation:** 2
**Contribution:** 2
**Rating:** 2
**Confidence:** 3

**Summary:**

This paper presents an Actor-Critic without Actor, a lightweight framework that removes the explicit actor network and directly generates actions from the gradient field of a noise-level critic.

**Strengths:**

This paper proposes an actor-critic algorithm without an actor and demonstrates its effectiveness by comparing it with several baselines across multiple tasks.

**Weaknesses:**

1. The paper lacks theoretical proofs for the proposed method, including: 1) Convergence analysis of the proposed method, and 2) Why the proposed method can outperform existing actor-critic methods and diffusion-based DRL methods.
2. How does the proposed method perform compared to actor-less DRL methods, such as the DQN series, on Atari tasks?
3. From Figures 5 and 6, the returns of the proposed method on Ant-v4, Walker2d-v4, Humanoid-v4, Swimmer-v4, Pusher-v4, and HalfCheetah-medium-v2 decrease below the baseline as learning progresses, showing no clear advantage.
4. The method should be compared with more recent SOTA actor-critic methods, including at least three standard actor-critic methods published in the last two years, besides SAC.
5. The diffusion-based DRL baseline is incomplete and should be compared with more baselines, such as "Ren A Z, Lidard J, Ankile L L, et al. Diffusion Policy Policy Optimization" (arXiv preprint arXiv:2409.00588, 2024).
6. Offline-to-online DRL needs validation on more tasks, such as Hopper, Ant, and Walker2d.
7. Besides comparing parameter counts, it is also important to compare actual hardware efficiency, such as wall-clock throughput, FLOPs, and VRAM usage, with more SOTA standard actor-critic algorithms and diffusion-based DRL methods.

**Questions:**

1. The paper lacks theoretical proofs for the proposed method, including: 1) Convergence analysis of the proposed method, and 2) Why the proposed method can outperform existing actor-critic methods and diffusion-based DRL methods.
2. How does the proposed method perform compared to actor-less DRL methods, such as the DQN series, on Atari tasks?
3. From Figures 5 and 6, the returns of the proposed method on Ant-v4, Walker2d-v4, Humanoid-v4, Swimmer-v4, Pusher-v4, and HalfCheetah-medium-v2 decrease below the baseline as learning progresses, showing no clear advantage.
4. The method should be compared with more recent SOTA actor-critic methods, including at least three standard actor-critic methods published in the last two years, besides SAC.
5. The diffusion-based DRL baseline is incomplete and should be compared with more baselines, such as "Ren A Z, Lidard J, Ankile L L, et al. Diffusion Policy Policy Optimization" (arXiv preprint arXiv:2409.00588, 2024).
6. Offline-to-online DRL needs validation on more tasks, such as Hopper, Ant, and Walker2d.
7. Besides comparing parameter counts, it is also important to compare actual hardware efficiency, such as wall-clock throughput, FLOPs, and VRAM usage, with more SOTA standard actor-critic algorithms and diffusion-based DRL methods.

---

### Official Review · Reviewer_GtS2 · 2025-10-30

**Soundness:** 2
**Presentation:** 2
**Contribution:** 1
**Rating:** 2
**Confidence:** 4

**Summary:**

This paper introduces a novel reinforcement learning (RL) framework called Actor-Critic without Actor (ACA). Traditional Actor-Critic (AC) methods rely on two separate networks—an actor and a critic. This separate design complicates training, is sensitive to hyperparameters, and increases computational and memory requirements, limiting its scalability. The ACA framework addresses these problems by completely removing the explicit actor network. Instead, it employs a "critic-guided denoising process": actions are generated by directly sampling from the gradient field of a "noise-level critic."

**Strengths:**

This paper eliminates the algorithmic and computational overhead of actor training and requires significantly fewer parameters than standard AC methods and diffusion-based methods.

ACA ensures that policy improvements (action generation) are always closely aligned with the SOTA value estimate of the critic, eliminating the "policy lag" found in traditional methods.

This framework retains the ability to capture diverse, multimodal behaviors (similar to diffusion models) without requiring a separate, computationally expensive actor network.

**Weaknesses:**

1.The method appears to share a significant conceptual foundation with existing work, particularly Q-Score Matching (QSM). As established, QSM's core idea is to train a diffusion model (the policy $\epsilon_{\theta}$) to match the Q-function's gradient field ($\nabla Q$). In contrast, the ACA method seems to primarily omit the gradient-matching step. Instead of training a separate $\epsilon_{\theta}$ to approximate this field, ACA directly utilizes the raw gradient of the critic, $\nabla_{a_t} Q_{\phi}(s, a_t, t)$, as the noise prediction $\hat{\epsilon}$. Given this, ACA appears to function less as a novel algorithmic paradigm and more as a simplification or a direct ablation of QSM.

2.The authors' repeated claims that the ACA framework is "lightweight"  and "parameter-efficient"  are highly misleading. The primary justification for this claim is the removal of the actor network, which reduces the total parameter count. This argument, however, ignores a critical trade-off: the method merely transfers the computational overhead from parameter storage (the actor network) to sampling latency (the iterative denoising process). In practical reinforcement learning applications, particularly in real-time control, inference latency (wall-clock time) is often a more severe bottleneck than the parameter count.

3.The entire theoretical foundation of the ACA framework, as articulated in Sec 3.1, rests upon a critical assumption: that its "critic-guided denoising process" (Definition 1)  can effectively sample from the target Boltzmann distribution $\hat{\pi} _ t \propto \exp(wQ _ {\phi})$ .However, the paper conspicuously fails to provide any theoretical guarantees regarding the convergence of this sampling process. The policy $\pi _ Q$ is an implicit policy, defined entirely by this sampler. It is well-established that such samplers, whether MCMC or diffusion-based, face significant challenges when sampling from complex, non-convex energy landscapes, which the Q-function $Q _ {\phi}$ represents. It is highly questionable whether the use of a finite and fixed number of steps, such as $T=20$, is sufficient to ensure the sampler converges to the stationary distribution. The authors should have provided an analysis to justify this choice, perhaps referencing empirical investigations similar to those in Fig 2 of the SDAC paper.

4.Considering the inference time, the non-converged performance on MuJoCo is insufficient to support the authors' SOTA claims. I need to see the performance metrics after training 3M iterations for each algorithm on different tasks until convergence.

5.This study lacks experimental comparison with the recent SOTA DIME, and I'd like to know the relationship between the Q-gradient field guidance proposed in the DACERv2 paper and yours? They seem very similar.

6.Could the author provide results comparing the time taken for single-step inference?

**Questions:**

See weakness.

---

### Official Review · Reviewer_VFZK · 2025-10-31

**Soundness:** 3
**Presentation:** 3
**Contribution:** 2
**Rating:** 4
**Confidence:** 4

**Summary:**

This paper proposes a new reinforcement learning framework, Actor-Critic without Actor (ACA). ACA eliminates the explicit actor network; instead, actions are generated directly from the gradient field of a noise-level critic, effectively transforming the critic into both a value estimator and an implicit policy generator. Inspired by diffusion model guidance, ACA formulates action sampling as a denoising process guided by $ \nabla_a Q(s, a, t) $. The critic is optimized with a two-term objective: (i) fitting terminal ($t=0$) temporal-difference targets, and (ii) regressing noisy-step values toward their denoised counterparts, thereby learning a noise-aware gradient field for action generation. Empirically, ACA demonstrates strong multi-modal coverage in a synthetic 2D bandit task and achieves competitive, sample-efficient performance across various MuJoCo benchmarks, while using significantly fewer parameters than diffusion-based actor–critic baselines.

**Strengths:**

The idea of removing the actor is conceptually elegant and effectively reduces both parameter and hyperparameter complexity. Training a critic that provides value estimates across different noise levels offers a principled way to stabilize the gradients used during denoising. This design also eliminates the “policy lag,” ensuring that the generated actions immediately reflect the most recent critic updates. Moreover, ACA naturally induces high-entropy, multi-modal behaviors without requiring explicit entropy regularization.

**Weaknesses:**

1.	While ACA motivates the Boltzmann-form policy $ \hat{\pi} \propto \exp(w Q) $ and draws intuition from classifier and diffusion guidance, the intermediate energy used in the framework serves as an approximate estimator. The paper lacks a quantitative analysis of the accuracy of this intermediate energy approximation, which is crucial for validating ACA, since implicit policy sampling directly relies on the gradient field of the noise-level critic. If the noise-level critic is not accurately learned, the resulting actions derived from its gradients may be unreliable.
2.	The connection between the trained noise-level critic, the denoising-based sampling dynamics, and overall policy improvement remains informal. The paper provides no convergence or consistency analysis for the full training loop.
3.	Since the actions used for Bellman backups are themselves generated by the critic through denoising, a feedback loop is formed. Although the paper introduces gradient normalization and target networks to mitigate instability, it does not analyze under what conditions this loop converges or fails.
4.	Iterative denoising in ACA is computationally more expensive than the single forward pass of an amortized actor. While the paper reports parameter savings, it only partially quantifies wall-clock training cost or action-generation throughput. The multi-step denoising process may hinder real-time deployment and large-batch training efficiency.

**Questions:**

1.	Does the critic-guided denoising policy $\pi_Q$ converge to the optimal policy under certain assumptions?
2.	Guidance weight (w) and diffusion steps (T) require manual tuning. Is there any adaptive scheduling ?
3.	Can ACA be accelerated with fewer denoising steps or learned step-size schedules?

---

### Note · Authors · 2025-11-13

**Comment:**

We sincerely appreciate the reviewers’ thoughtful and constructive feedback. Because the experiment was conducted under a tight schedule, we made an error that affected part of our paper. For this reason, we have decided to withdraw the submission.

We will carefully incorporate the reviewers’ comments, address the identified issue, and prepare a more improved and reliable version of the paper. We sincerely appreciate your time and understanding.

**Withdrawal Confirmation:**

I have read and agree with the venue's withdrawal policy on behalf of myself and my co-authors.